# Effects of Litter Decomposition on Soil N in *Picea mongolica* Forest at Different Forest Ages

**Yunchao Liu** [1,2], **Lixin Chen** [1,*], **Wenbiao Duan** [1], **Yongan Bai** [3] **and Xiaolan Li** [2]

1   College of Forestry, Northeast Forestry University, Harbin 150040, China; liuyunchao616@126.com (Y.L.); dwbiao@163.com (W.D.)
2   School of Resource and Environmental Sciences, Chifeng University, Chifeng 024000, China; lixiaolan97@163.com
3   Baiyinaobao National Nature Reserve Management Administration, Baiyinaobao, Chifeng 025350, China; bya1982@163.com
*   Correspondence: lxchen88@163.com

**Abstract:** In order to study the effects of litter decomposition on soil nitrogen of *Picea mongolica* in different forest ages, young forest (0–5a), middle-aged forest (5–30a), and near-mature forest (30–40a) stands were selected in the Baiyinaobao National Nature Reserve. Litter decomposition was assessed using the decomposition bag method. The seasonal and vertical spatial variation characteristics of total N, $NH_4^+$—N, and $NO_3^-$—N caused by litter decomposition in *P. mongolica* forest soil were studied for different stand ages. Results showed that: (1) There was a positive correlation between litter N content and soil organic matter, total N content, and $NO_3^-$—N content across different forest ages ($p < 0.05$). There was a negative correlation between litter N and $NH_4^+$—N contents. A negative correlation between litter C content and soil organic matter, total N, and $NO_3^-$—N contents was also observed. (2) In this study, the total N and $NO_3^-$—N increased with the increase in N content during litter decomposition. $NH_4^+$—N in the soil was positively correlated with sample date, soil $NO_3^-$—N, and forest age ($p < 0.05$), and negatively correlated with soil depth ($p < 0.01$). $NO_3^-$—N in the soil was negatively correlated with sample date and forest age ($p < 0.05$), and significantly negatively correlated with soil depth ($p < 0.01$). (3) the $NH_4^+$—N content is greater than that of $NO_3^-$—N in each soil layer for the three forest ages. The correlation analysis indicated which factors influenced $NH_4^+$—N and $NO_3^-$—N in the soil. The content decreased during February and November and increased in May and August. (4) The total N, $NH_4^+$—N, and $NO_3^-$—N in the forest soils across the three forest ages increased with the depth of the soil layer (0–50 cm) and showed an overall downward trend. The contents of $NH_4^+$—N in the soil layer from the young forest (0–10 cm, 10–20 cm and 20–30 cm, 30–40 cm, and 40–50 cm) differed significantly ($p < 0.05$), as did the $NO_3^-$—N results ($p < 0.05$), while results from the middle-aged forest and near-mature forest increased with soil layer depth. There was no significant difference in the $NH_4^+$—N soil content. (5) The $NH_4^+$—N in the forest soils showed a trend from mature forest > middle-aged forest > young forest. This trend for soil $NO_3^-$—N content is consistent with that of the $NH_4^+$—N content in the *Picea mongolica* forest soil.

**Keywords:** horqin sandy land; *Picea mongolica* forest; $NH_4^+$—N; $NO_3^-$—N; seasonal dynamics; litter decomposition

## 1. Introduction

N is a major macronutrient necessary for plant growth and development, as well as the main limiting factor for forest ecosystem productivity [1,2]. $NH_4^+$—N and $NO_3^-$—N are available N that can be directly absorbed and utilized by plants. Changes in their content in the soil directly affect the migration and transformation of soil N and plant productivity [3]. The N, $NH_4^+$—N, and $NO_3^-$—N contents in the soil of different forest types and surfaces have been found to differ significantly. Many studies have been undertaken on the chemometric characteristics of soil N in forest ecosystems in China and

other countries. These include different forest ages [4,5], land use modes [6,7], succession stages [8] altitudes [9,10], tree species [11,12], and more. In recent years, several studies have been conducted on different tree species, as different species have different habitats with different soil physical and chemical properties, soil nutrient composition and distribution within the ecosystem [13,14]. *Picea mongolica* forest is a rare temperate coniferous species in China, which is mainly distributed in Keshketeng county on the eastern edge of Hunshandak Sandy Land in China. This area has the characteristics of horizontal spatial richness and vertical spatial heterogeneity of grassland in the farming–pastoral transitional zone in north China. However, few studies have been conducted in the *P. mongolica* forest at the North China forest ecotone. Therefore, it is of great significance to study the characteristics of seasonal variation in soil N in this forest.

*P. mongolica* is a tree species native to the northern agropastoral ecotone. This species mainly grows on sunny slopes in the mountains at an altitude of 1300–1500 m. To date, few studies have been conducted on the quantitative characteristics of soil N in this ecosystem with a focus on forest age, soil depth, and seasonality. In this study, analyses of variations in forest age, soil depth, and seasonal dynamic changes are combined. We compared and analyzed seasonal dynamic changes in soils at different levels of *P. mongolica* forest and examined the migration and transformation tendency for total and available N to provide a scientific basis for the management of the *P. mongolica* forests. The results from this study can provide a reference for use in the management and cultivation of artificial forests.

## 2. Materials and Methods

### 2.1. Site Description

The study area is located in the BaiyinOboo Nature Reserve (43°30′–43°36′ N, 117°03′–117°16′ E) in Keshketeng county on the eastern edge of Hunshandak Sandy Land in China, with an area of approximately 1947 km². The region has a temperate grassland climate, with an average annual temperature of −1.4 °C, an average temperature of −23.4 °C in January, and 17.4 °C in July. The average annual frost-free period is 78 days, with an annual precipitation of 360–440 mm, an average annual evaporation of 1035.6 mm, and an altitude of 1300–1500 m. The soil type is gray forest soil, and the flora belongs to the plant distribution area of Mongolia. In terms of plant species composition, common taxa that are present include *P. mongolica*, *Larix decidua* Mill., *Juniperussabina* var. *davurica* (Pallas) Farjon, *Spiraea aquilegifolia*, *S. dahurica* (Rupr.) Maxim., and *Pinus tabuliformis* Carr.

### 2.2. Sample Plot Setting and Collection and Pre-Treatment of Litter

Based on the initial field investigations conducted in early August 2016, three forest areas with young (0–5a), middle-aged (5–30a), and near-mature (30–40a) *P. mongolica* forest were selected in the Baiyinaobao National Nature Reserve. In each forest stand, three 10 × 10 m² sampling plots that were in full sun, uniformly distributed, and with good vegetation growth were selected. Five 1 × 1 m² small quadrats were chosen in an "s" shape within each sample plot. There were 15 small plots in each forest stand. The key vegetation parameters of the sampling plots are shown in Table 1 and monthly average temperature and monthly precipitation in different months is shown in Table 2.

**Table 1.** Key vegetation parameters of the sampling plots. The data presented here were collected during March 2017. Values presented are the mean ± standard error. Soil sample numbers were similar to 15(There are three plots for each forest age, and five samples are taken from each plot according to the s-shape).

| | Forest Type | | | Soil Layer (0–30 cm) | | |
|---|---|---|---|---|---|---|
| Age of Stand (a) | Elevation Gradient m | Average DBH cm | Average Height m | Soil pH | Soil Moisture/ % | Soil Temperature °C |
| Young forest (0–5a) | 1344 | 1.79 ± 2.15 | 2.06 ± 0.71 | 7.01 ± 0.12 | 35.95 ± 0.45 | −15.78 ± 2.49 |
| Middle-aged forest (5–30a) | 1352 | 8.03 ± 4.81 | 8.15 ± 1.86 | 6.67 ± 0.03 | 42.45 ± 4.13 | −14.59 ± 2.61 |
| Near-mature forest (30–40a) | 1342 | 14.26 ± 4.12 | 13.94 ± 2.15 | 6.69 ± 0.01 | 17.28 ± 2.61 | −15.33 ± 2.42 |

**Table 2.** Monthly average temperature, monthly precipitation in different months, and soil pH value in young forest (Y-F); middle-aged forest (M-F); and near-mature forest (N-F).

| Months | Monthly Average Temperature (°C) | Monthly Precipitation (mm) | Soil pH Value | | |
|---|---|---|---|---|---|
| | | | Y-F | M-F | N-F |
| 17 March | −10.4 | 1.38 | 7.28 | 6.72 | 6.97 |
| 17 May | 14.5 | 39.91 | 7.06 | 6.72 | 6.79 |
| 17 July | 22.62 | 95.12 | 6.37 | 5.89 | 6.49 |
| 17 September | 13.16 | 59.11 | 6.95 | 6.54 | 6.63 |
| 17 November | −6.48 | 3.93 | 7.14 | 6.26 | 6.54 |

During the period of maximum litter (mainly leaf litter) of *P. mongolia* in late October 2016, freshly fallen needles (hereafter referred to as "litter") were collected, placed in airtight bags, and immediately transported to the laboratory. Needles collected from 45 small quadrats of nine plots in three stands were dried at 80 °C until a constant weight was obtained. The litter of three stands having the same mass (1000 g) was completely mixed. Some litter was crushed and screened through a 60 mesh to analyze the initial C, and N contents in the litter of *P. mongolia* forests.

After drying, the mixed litter (20 g, error less than 0.01 g) was placed into a 15 cm × 15 cm decomposition bag made of 0.15 mm nylon gauze. In November 2016, after removing the surface litter from the sample plots, the decomposition bags containing the litter were returned to the nine sample plots of the three stands. The decomposition bags were placed parallel to the sample plots without overlapping. The litter was kept flat in the net bags to ensure complete contact with the humus layer and as close to the natural decomposition state as possible. Nine litter bags were placed in each sample plot; thus, in total, 27 litter bags were placed in each stand. Samples were collected regularly in March, May, July, September, and November 2017 (as some areas were still covered by snow before March, sample collection prior to March was not conducted). During these months, a litter decomposition bag was retrieved from each of the three plots in each stand, nine bags were collected at a time, and gravel, roots, and plant and animal debris were removed. The litter from the bags in each stand was thoroughly mixed and transported back to the laboratory for further analysis. Subsequently, the litter was continuously dried at 80 °C to obtain constant weight and litter retention. Meanwhile, the litter samples were collected from the three forest types, and the soil under the litter bags of 0~10 cm, 10~20 cm, 20~30 cm, 30~40 cm, and 40~50 cm was collected for indoor treatment and analysis.

*2.3. Sample Analysis and Data Statistics*

The litter samples retrieved from the same field in each stand were dried to a constant weight at 80 °C, then mixed evenly, crushed, sifted through 60 mesh, and put into ziplocked bags for testing. The litter measurement indexes were total C and total N. Total C of litter was determined using an SSM-TOC analyzer (Shanghai Meta-analysis, Shanghai, China, TOC-L). Total nitrogen (TN) was determined by sulfuric acid-perchloric acid

elimination cooking and the Kjeldahl method (Kjeldahl nitrogen meter, Beijing, China, San-pinkechuang, Spd60). Following air drying of the samples indoors, impurity removal, and grinding through a 100-mesh screen, the soilpH value, total N, $NH_4^+$—N, and $NO_3^-$—N were determined. The pH value of the soil was determined by acidity meter after aqueous solution extraction (soil–water ratio: 1:2.5) (Kcidity meter, Thunder magnetic, Shanghai, China, PHS-2F), total N was determined using $H_2SO_4$-$H_2O_2$ digestion indophenol blue colorimetry. $NH_4^+$—N was determined using indophenol blue colorimetry (UV-Vis Spectrophotometer, Beijing, China, TU-1950) [15]. $NO_3^-$—N was determined using phenol disulfonic acid colorimetry in China National Standard (determination of $NO_3^-$—Nin the forest soils) (ly/y1233-1999).

Excel 2017 and SPSS 22.0 software were used for statistical analysis of data. One-way analysis of variance (ANOVA) and least significant difference (LSD) were used for analysis of variance and multiple comparisons ($\alpha = 0.05$). The Pearson method was used for correlation analysis. Amos 22.0 software was used to establish a structural equation model. The data in Table 3 is the mean $\pm$ standard deviation.

**Table 3.** Nutrient content of *P. mongolica* forest in different stand ages and sampling times in young forest (Y-F); middle-aged forest (M-F); and near-mature forest (N-F). Different uppercase letters in the same column represent the same factor and significant difference between different ages of different stands ($p < 0.05$), and different lowercase letters in the same line represent significant difference between different factors of the same index ($p < 0.05$). The same below.

| Sampling Time | Forest Type | Total C (mg·kg$^{-1}$) | Total N (mg·kg$^{-1}$) | C/N |
|---|---|---|---|---|
| 15 November 2016 | Litters | 262.74 | 4.29 | 61.21 |
| 15 March 2017 | Y-F | 261.96 $\pm$ 0.06aA | 4.33 $\pm$ 0.01aB | 60.49 $\pm$ 0.3aA |
| | M-F | 257.69 $\pm$ 0.35aA | 4.37 $\pm$ 0.02aA | 58.97 $\pm$ 0.1aA |
| | N-F | 254.88 $\pm$ 0.13aB | 4.43 $\pm$ 0.01aB | 57.55 $\pm$ 0.10aA |
| 17 May 2017 | Y-F | 246.99 $\pm$ 0.02bA | 4.55 $\pm$ 0.02bA | 54.27 $\pm$ 0.01bA |
| | M-F | 239.13 $\pm$ 0.06bB | 4.68 $\pm$ 0.01bA | 51.10 $\pm$ 0.02bA |
| | N-F | 236.01 $\pm$ 0.42bB | 4.69 $\pm$ 0.05bA | 50.32 $\pm$ 0.47ba |
| 15 July 2017 | Y-F | 233.49 $\pm$ 4.09bA | 4.77 $\pm$ 0.01bA | 48.95 $\pm$ 0.13bA |
| | M-F | 226.46 $\pm$ 5.53bA | 4.84 $\pm$ 0.01bA | 46.79 $\pm$ 0.74bA |
| | N-F | 224.35 $\pm$ 16.09bA | 4.98 $\pm$ 0.01bA | 45.05 $\pm$ 1.11ba |
| 18 September 2017 | Y-F | 228.59 $\pm$ 1.86bA | 5.09 $\pm$ 0.01bA | 44.91 $\pm$ 3.25cA |
| | M-F | 204.22 $\pm$ 1.86bA | 5.37 $\pm$ 0.04bA | 38.03 $\pm$ 0.34ca |
| | N-F | 198.81 $\pm$ 1.36bA | 5.35 $\pm$ 0.04bA | 37.16 $\pm$ 0.54ca |
| 15 November 2017 | Y-F | 203.12 $\pm$ 0.79aA | 5.28 $\pm$ 0.01aA | 38.47 $\pm$ 0.15cA |
| | M-F | 193.43 $\pm$ 2.59aA | 5.42 $\pm$ 0.02aA | 35.69 $\pm$ 0.42cA |
| | N-F | 187.28 $\pm$ 1.89aA | 5.28 $\pm$ 0.07aA | 35.47 $\pm$ 0.16cA |

## 3. Results and Analysis

### 3.1. Chemical Composition of P. mongolica Forest Litter in Different Stands

The nutrient content of litter can be used to measure its quality [16].

As shown in Table 3, during the decomposition of *P. mongolica* litter, the C and C/N contents in leaf litter of the young, middle-aged, and near-mature forests showed a decreasing trend with time. The C content of leaf litter of the three forest types was from March to November 2017. It decreased by 9.79%, 8.29%, and 7.04%, respectively. In March, there was no significant difference in the C content of litterfall among the three forest ages ($p > 0.05$), while in May, July, and September, there was significant difference in the C content of litterfall among the three forest ages ($p < 0.05$). The C/N of the three forest types were from March to November 2017, it decreased by 4.86%, 3.94%, and 3.84%, respectively. In 2017, the C/N contents of litterfall of three forest ages were significantly different between young forest, middle forest age, and near mature forest ($p < 0.05$), but not between middle forest age and near mature forest ($p > 0.05$). However, the N content of leaf litter in the young, middle-aged, and near-mature forests increased by 17.99%, 19.34%, and 16.21% from March to November in 2017, respectively. In March, there was no significant difference in N content of litterfall among the three forest ages ($p > 0.05$), while in May and September, there was a significant difference in N content between young forest, middle forest age,

and near mature forest ($p < 0.05$), and in July, middle forest age, near mature forest, and young forest ($p < 0.05$). There were significant differences in the age of middle forest, young forest, and near mature forest in November.

### 3.2. Seasonal Variation of Soil Total N Content with Soil Depth and Stand Age

During March and November, the total N content of the soils from *P. mongolica* forest stands across the three age groups showed a downward trend with increasing soil depth (Table 4, Figure 1). There were significant differences in the total N content between 0–10 cm, 10–20 cm, 20–30 cm, 30–40 cm, and 40–50 cm soil layers ($p < 0.05$). During March and November, the maximum values of total N across the three stand ages were recorded from the 0–20 cm soil layer. The minimum values were recorded from the 30–50 cm soil layer. The average soil total N contents recorded in near-mature, young, and middle-aged forests were 18.32 mg/kg, 17.87 mg/kg, and 15.96 mg/kg, respectively. During November, the average soil total N contents recorded in young, near-mature, and middle-aged forests in this period were 23.04 mg/kg, 18.01 mg/kg, and 15.72 mg/kg, respectively.

**Table 4.** Soil ammonium nitrogen and nitrate nitrogen contents and their vertical distribution in different stand ages and times in young forest (Y-F); middle-aged forest (M-F); and near-mature forest (N-F). The sampling time is 2017. Different lowercase letters in the same column denote significant differences between soil layers. Different capital letters in the same row denote significant differences across altitudes at the 0.05 level.

| Times | Soil Layer (cm) | Total N (mg·kg⁻¹) | | | NNH₄N (mg·kg⁻¹) | | | NNO₃N (mg·kg⁻¹) | | |
|---|---|---|---|---|---|---|---|---|---|---|
| | | Y-F | M-F | N-F | Y-F | M-F | N-F | Y-F | M-F | N-F |
| 15 March | 0–10 | 37.01 ± 1.43bC | 22.73 ± 0.66cA | 26.88 ± 0.23bB | 0.26 ± 0.16dB | 0.11 ± 0.14cA | 0.08 ± 0.02cA | 21.93 ± 11.75cA | 24.03 ± 10.39cA | 25.19 ± 3.29cA |
| | 10–20 | 14.92 ± 0.64aA | 13.82 ± 0.29aA | 30.25 ± 0.47bB | 0.13 ± 0.06cC | 0.07 ± 0.04bB | 0.03 ± 0.03bA | 12.81 ± 3.01bA | 11.87 ± 1.63bA | 15.47 ± 10.85bB |
| | 20–30 | 12.91 ± 0.52aa | 10.46 ± 0.83aA | 10.38 ± 0.55aA | 0.06 ± 0.02bB | 0.03 ± 0.02aA | 0.03 ± 0.03bA | 9.41 ± 0.58aA | 8.91 ± 0.56aA | 9.98 ± 1.80aA |
| | 30–40 | 12.52 ± 0.81aA | 15.96 ± 0.00bB | 12.96 ± 0.06aA | 0.04 ± 0.01aB | 0.03 ± 0.00aA | 0.03 ± 0.00bA | 8.69 ± 0.54aA | 12.86 ± 1.54bB | 9.89 ± 1.12aA |
| | 40–50 | 12.02 ± 0.41aA | 12.55 ± 0.00aA | 11.13 ± 0.64aA | 0.04 ± 0.01aB | 0.05 ± 0.03aB | 0.02 ± 0.01aA | 10.37 ± 1.08aA | 10.25 ± 1.33aA | 10.61 ± 3.04aA |
| 17 May | 0–10 | 59.95 ± 1.21bC | 30.32 ± 1.44bB | 25.19 ± 0.76aA | 0.18 ± 0.21cB | 0.09 ± 0.04cA | 0.11 ± 0.09bA | 33.57 ± 14.81bB | 22.59 ± 6.16aA | 36.09 ± 12.77cB |
| | 10–20 | 62.42 ± 0.64bB | 33.03 ± 1.11bA | 41.75 ± 1.21bA | 0.06 ± 0.01aA | 0.06 ± 0.03bA | 0.13 ± 0.17bB | 38.12 ± 16.38bB | 34.47 ± 15.62bA | 42.67 ± 24.25dC |
| | 20–30 | 22.92 ± 1.01aA | 43.37 ± 1.96cB | 27.42 ± 0.92aA | 0.08 ± 0.08bB | 0.11 ± 0.12cC | 0.02 ± 0.01cA | 21.01 ± 1.64aA | 36.09 ± 9.98bB | 40.24 ± 15.46dB |
| | 30–40 | 21.23 ± 1.25aA | 21.75 ± 0.35aA | 23.97 ± 0.62aA | 0.13 ± 0.13bA | 0.02 ± 0.02aB | 0.02 ± 0.01aA | 22.47 ± 2.84aA | 30.61 ± 18.72bB | 31.86 ± 14.58bB |
| | 40–50 | 23.25 ± 0.87aA | 26.08 ± 1.67aA | 25.32 ± 0.98aA | 0.07 ± 0.05aB | 0.02 ± 0.01aA | 0.02 ± 0.01aA | 20.05 ± 0.54aA | 21.96 ± 2.85aA | 24.12 ± 1.77aA |
| 15 July | 0–10 | 59.55 ± 0.52bC | 31.42 ± 0.46bB | 25.79 ± 0.27aA | 0.06 ± 0.01aB | 0.12 ± 0.01cA | 0.74 ± 0.02dB | 54.39 ± 0.01bC | 31.25 ± 0.01bB | 22.42 ± 0.01aA |
| | 10–20 | 62.45 ± 0.69bC | 33.34 ± 0.58cA | 41.16 ± 0.20bB | 0.06 ± 0.01aA | 0.07 ± 0.00bB | 0.06 ± 0.01cA | 59.95 ± 0.01bB | 33.41 ± 0.02bA | 36.19 ± 0.01bA |
| | 20–30 | 23.82 ± 0.55aA | 44.43 ± 0.12dB | 26.99 ± 0.17aA | 0.19 ± 0.01bA | 0.33 ± 0.01bB | 0.03 ± 0.01bB | 21.15 ± 0.02aA | 42.59 ± 0.01cB | 23.49 ± 0.03aA |
| | 30–40 | 22.15 ± 0.35aA | 21.57 ± 0.04aA | 24.6 ± 0.48aA | 0.31 ± 0.01cA | 0.03 ± 0.01aA | 0.03 ± 0.01bA | 20.55 ± 0.01aA | 21.19 ± 0.00aA | 22.31 ± 0.01aA |
| | 40–50 | 23.90 ± 0.27aA | 27.28 ± 0.40bA | 25.60 ± 0.50aA | 0.15 ± 0.01bA | 0.03 ± 0.01aB | 0.03 ± 0.01aB | 19.99 ± 0.01aA | 20.66 ± 0.01aA | 21.49 ± 0.01aA |
| 18 September | 0–10 | 23.75 ± 0.35aA | 51.03 ± 0.52bC | 38.51 ± 0.28bB | 0.01 ± 0.02aA | 0.08 ± 0.08cB | 0.06 ± 0.07bB | 25.97 ± 5.34aA | 42.31 ± 16.01cB | 27.90 ± 8.46aA |
| | 10–20 | 27.13 ± 0.65aA | 63.61 ± 0.01cC | 47.61 ± 0.30bB | 0.02 ± 0.03aA | 0.02 ± 0.02aA | 0.08 ± 0.06cB | 33.05 ± 9.34bA | 39.16 ± 17.72cB | 37.72 ± 12.24cB |
| | 20–30 | 25.06 ± 0.10aA | 50.955 ± 0.02bB | 54.35 ± 0.13cB | 0.03 ± 0.04bA | 0.11 ± 0.08dB | 0.06 ± 0.04bA | 22.92 ± 1.90aA | 36.99 ± 12.81bB | 38.53 ± 18.15cB |
| | 30–40 | 41.26 ± 0.15cB | 34.39 ± 0.01aA | 25.35 ± 0.29aA | 0.02 ± 0.03aA | 0.04 ± 0.03aB | 0.07 ± 0.11bC | 31.85 ± 7.71bA | 31.59 ± 8.13aA | 33.12 ± 8.73bA |
| | 40–50 | 31.13 ± 0.11bA | 33.14 ± 0.03aA | 25.43 ± 0.05aA | 0.04 ± 0.05bA | 0.07 ± 0.03aB | 0.04 ± 0.06aA | 33.08 ± 3.51bB | 34.20 ± 6.24cB | 24.84 ± 9.97aA |
| 15 November | 0–10 | 30.76 ± 0.56cB | 12.91 ± 0.22aA | 21.34 ± 0.42aA | 0.05 ± 0.01cB | 0.03 ± 0.04aA | 0.11 ± 0.07cC | 17.25 ± 10.92bA | 20.69 ± 9.41bB | 26.82 ± 5.85bC |
| | 10–20 | 17.08 ± 0.30aA | 22.36 ± 0.37bB | 28.00 ± 0.29bC | 0.04 ± 0.01cA | 0.05 ± 0.02bA | 0.09 ± 0.04bB | 9.62 ± 4.24aA | 15.11 ± 4.80aA | 35.73 ± 9.73cB |
| | 20–30 | 14.14 ± 0.15aA | 21.05 ± 0.17bB | 13.94 ± 0.36aA | 0.02 ± 0.01aA | 0.05 ± 0.01bB | 0.10 ± 0.06cC | 11.90 ± 1.79aA | 14.55 ± 5.40aA | 11.24 ± 1.95aA |
| | 30–40 | 29.39 ± 0.54bB | 11.07 ± 0.24aA | 9.68 ± 0.32aA | 0.03 ± 0.02bA | 0.05 ± 0.03cB | 0.10 ± 0.02bC | 16.79 ± 10.23bA | 11.51 ± 6.26aA | 13.31 ± 4.25aA |
| | 40–50 | 23.84 ± 0.33bB | 11.23 ± 0.28aA | 17.13 ± 0.26bB | 0.03 ± 0.01aA | 0.03 ± 0.01aA | 0.07 ± 0.04aB | 15.59 ± 5.06bA | 12.32 ± 1.65aA | 27.18 ± 25.29bB |

By recording the month, soil depth, forest type, pH value, soil temperature, soil humidity, soil organic matter, litter N content, litter carbon content, and litter C/N content as independent variables, a fitted linear regression model for all N was constructed. From the fitted linear results in Table 5, there was a good linear fit for the model with a significant *p* value of 0.001. The results are shown in Table 4. The total N content across the three forest ages from May, July, and September first increased and then decreased with increasing soil depth. There was a significant difference in the total N content between 0–10 cm, 10–20 cm, 20–30 cm, 30–40 cm, and 40–50 cm ($p < 0.05$) during May. The maximum age total N contents for young and near-mature forests were recorded from the 10–20 cm soil layer. The maximum age total N content from the middle-aged forest was recorded from the 20–30 cm soil layer. The minimum forest age was recorded from the 30–50 cm soil layer. The average soil total N contents in the young, middle-aged, and near-mature forests were 37.95 mg/kg, 30.96 mg/kg, and 28.73 mg/kg, respectively. In July, the maximum value of total N in young and near-mature forests was recorded from the 10–20 cm soil layer. The maximum total N content from the middle-aged forests was recorded from the 20–30 cm soil layer. The minimum value of total N from three forest age groups was recorded from the 30–50 cm soil layer. The average soil total N contents in the young, middle-aged, and near-mature forests were 38.37 mg/kg, 31.61 mg/kg, and 28.82 mg/kg, respectively. In September, the

maximum values of total N from the young and the middle-aged forests were recorded from the 10–20 cm soil layer. The maximum value of total N from the near-mature forest was recorded from the 20–30 cm soil layer. The minimum value across the three forest age groups was recorded in the 30–50 cm soil layer. The average soil total N content recorded in the middle-aged, near-mature, and young forests were 46.62 mg/kg, 38.24 mg/kg, and 29.66 mg/kg, respectively. In 2017, the total N content across the three forest age groups first increased and then decreased over time. In May and July, the highest total N content was recorded across all three forest age groups.

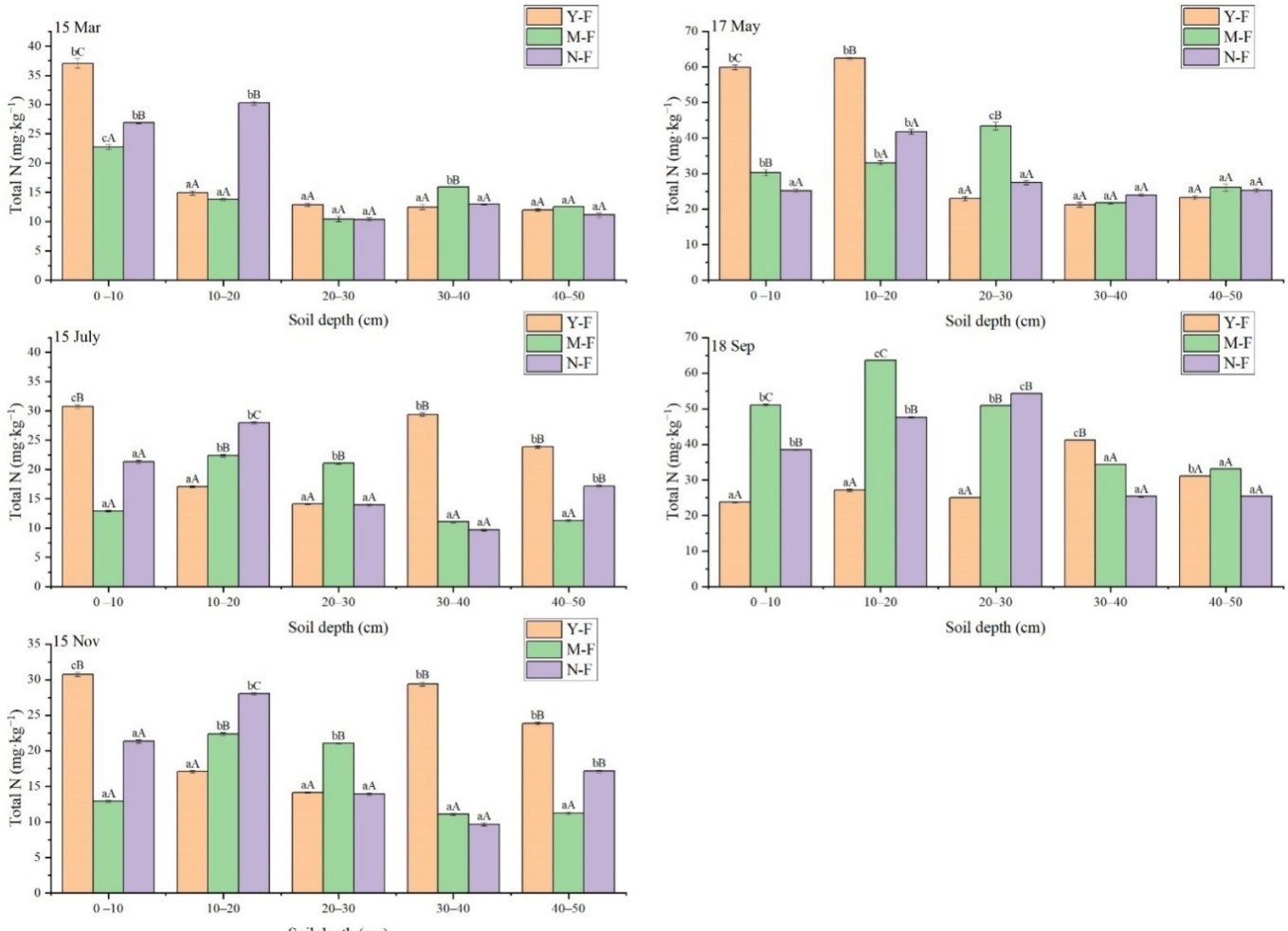

**Figure 1.** Plot of soil total nitrogen variation with soil depth. Lowercase letters represent the significant difference of different soil layers in the same forest age, and uppercase letters represent the significant difference of different forest ages in the same soil layer.

**Table 5.** Variance analysis of soil total nitrogen in *P. mongolica* forest (*F*-test). The dependent variable was total N.

| Model | | Sum of Squares | Freedom | Mean Square | *F* | Significance |
|---|---|---|---|---|---|---|
| 1 | Regression | 632,256,641 | 10 | 63,225,664 | 2.319 | 0.013 |
| | Residual | 7,062,498,135 | 259 | 27,268,333 | | |
| | Total | 7,694,754,776 | 269 | | | |

By recording the sampling time, soil depth, forest type, pH value, soil temperature, soil humidity, soil organic matter, litter N content, litter carbon content, and litter C/N content as independent variables, the fitted linear regression model for all n was constructed. From

the fitted linear results in Table 5, the regression value hasa significant *p*-value of 0.013. The results are shown in Table 6.

**Table 6.** Multivariate analysis of variance of soil total nitrogen of *P. mongolica* forest (*t*-test).

| Model | Nonstandard Coefficient | | Standardization Coefficient | *t* | Significance |
|---|---|---|---|---|---|
| (Constant) | 52,347.16 | 27,613.11 | | 1.90 | 0.06 |
| Particular year | −3899.58 | 1154.61 | −1.25 | −3.38 | 0.01 |
| Forest type | −1327.88 | 550.65 | −0.20 | −2.41 | 0.02 |
| Soil depth | 5.12 | 24.72 | 0.01 | 0.21 | 0.08 |
| pH value | −118.22 | 1006.91 | −0.01 | −0.12 | 0.19 |
| Humidity | −367.86 | 390.42 | −0.11 | −0.94 | 0.03 |
| Temperature | −24.02 | 78.97 | −0.05 | −0.30 | 0.04 |
| Litter N content | −3047.91 | 3006.83 | −0.22 | −1.01 | 0.31 |
| Litter C content | 255.93 | 165.93 | 1.21 | 1.54 | 0.12 |
| Litter CN ratio | −1617.78 | 569.09 | −2.82 | −2.84 | 0.01 |
| Soil organic matter content | 8.19 | 6.66 | 0.09 | 1.23 | 0.02 |

From the fitted linear results in Table 6, humidity, temperature, and CN ratio of litter and organic matter have had a considerable impact on the level of total N, and the highest contribution rate. Although, the impact of litter N content and pH value is slightly lower, the *p*-values for all the other influencing factors, except for the above factors, were greater than 0.05, and the results were not significant.

### 3.3. Characteristics of Seasonal Variation of Soil $NH_4^+$—N Content with Soil Depth and Stand Age

In March and November, the content of $NH_4^+$—N in soil for the three forest standages showed an overall downward trend with increasing soil depth (Table 4 and Figure 2). There were significant differences in the $NH_4^+$—N content in the 0–10 cm, 10–20 cm, 20–30 cm, 30–40 cm, and the 40–50 cm soil layers ($p<0.05$).In March, the maximum values of $NH_4^+$—N across the three forest ages were recorded from the 0–10 cm soil layer, and the minimum values were recorded in the 30–50 cm soil layer. The average soil $NH_4^+$—N contents of young, middle-aged, and near-mature forests were 0.11 mg/kg, 0.07 mg/kg, and 0.04 mg/kg, respectively. In November, the maximum values of $NH_4^+$—N across the three forest ages were recorded from the 0–20 cm soil layer, and the minimum values were recorded from the 30–50 cm soil layer. The average soil $NH_4^+$—N content recorded from near-mature, middle-aged, and young forests was0.09 mg/kg, 0.04 mg/kg, and 0.03 mg/kg, respectively. The $NH_4^+$—N content across the three forest ages in May, July, and September first increased and then decreased with increasing soil depth. The soil total N contents between the 0–10 cm, 10–20 cm, 20–30 cm, 30–40 cm, and 40–50 cm soil layers differed significantly ($p<0.05$). In May, the maximum value of $NH_4^+$—N for young and near-mature forests was recorded from the 10–20 cm soil layer. The maximum value of total N for middle-aged forest was recorded from the 20–30 cm soil layer. The minimum value of $NH_4^+$—N for the three forest age groups was recorded from the 30–50 cm soil layer. The average soil $NH_4^+$—N content recorded from young, middle-aged, and near-mature forests were 0.10 mg/kg, 0.06 mg/kg, and 0.06 mg/kg, respectively. In July, the maximum values of $NH_4^+$—N for young and near-mature forests were recorded from the 10–20 cm soil layer. The maximum value of $NH_4^+$—N from the middle-aged forest was recorded from the 20–30 cm soil layer. The minimum value of $NH_4^+$—N for the three forest age groups was recorded from the 30–50 cm soil layer. The average soil $NH_4^+$—N contents recorded for near-mature, young, and middle-aged forests were 0.18 mg/kg, 0.15 mg/kg, and 0.12 mg/kg, respectively. In September, the maximum values of $NH_4^+$—N for young and near-mature forests were recorded from the 10–20 cm soil layer. The maximum value of $NH_4^+$—N from the middle-aged forest was recorded from the 20–30 cm soil layer. The minimum value of $NH_4^+$—N for the three forest age groups was recorded from the 30–50 cm soil layer. The average soil total N contents recorded for middle-aged, near-mature, and young forests were 0.06 mg/kg, 0.06 mg/kg, and 0.04 mg/kg, respectively.

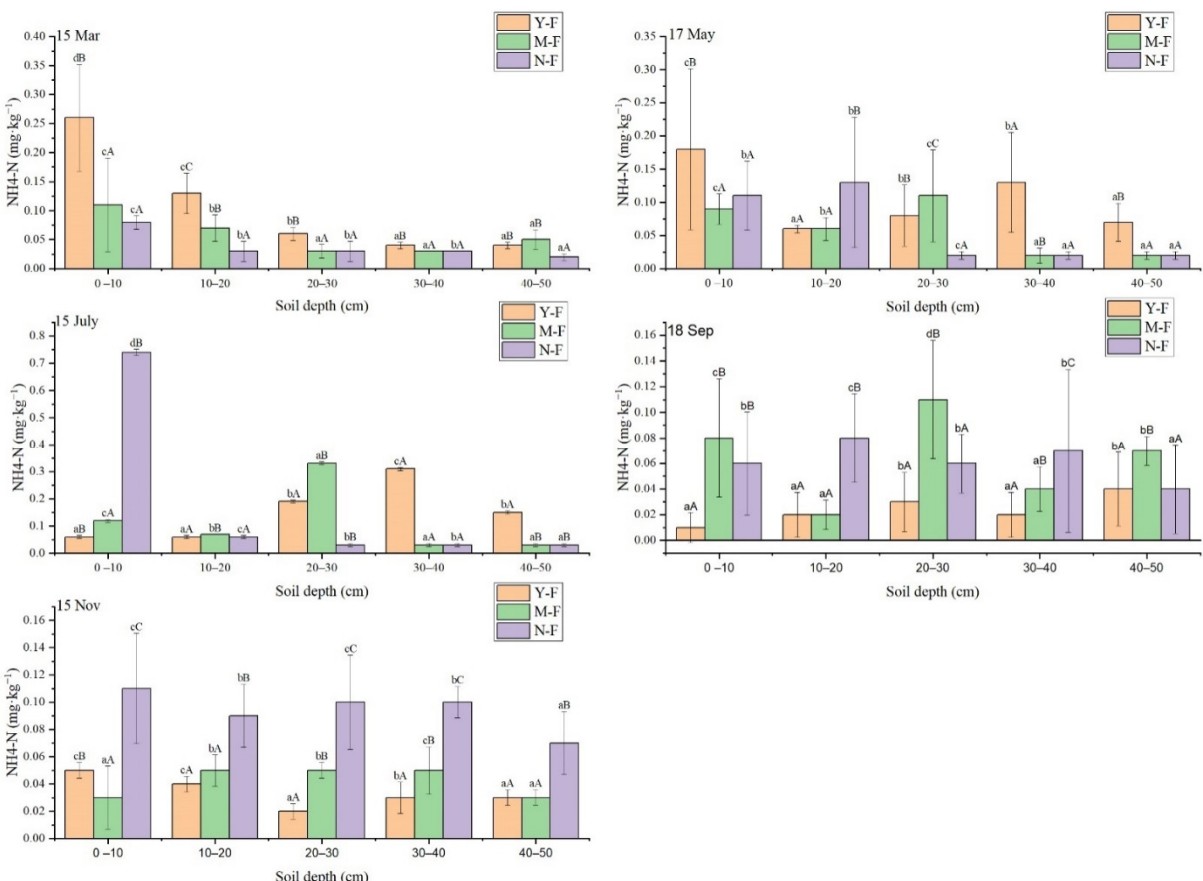

**Figure 2.** Plot of soil $NH_4^+$—N variation with soil depth. Lowercase letters represent the significant difference of different soil layers in the same forest age, and uppercase letters represent the significant difference of different forest ages in the same soil layer.

By recording sampling time, soil depth, forest type, pH value, soil temperature, soil humidity, soil organic matter, litter N content, litter carbon content, and litter C/N content as independent variables, the fitted linear regression model for all n was constructed. From the fitted linear results in Table 7, the fitted linear results were good. There was a significant *p*-value of 0.001.The results are shown in Table 8.

**Table 7.** Variance analysis of soil ammonium nitrogen in *P. mongolica* forest (*F*-test).

| Model | | Sum of Squares | Freedom | Mean Square | F | Significance |
|---|---|---|---|---|---|---|
| 1 | Regression | 0.681 | 10 | 0.068 | 8.488 | 0.001 |
| | Residual | 2.079 | 259 | 0.008 | | |
| | Total | 2.761 | 269 | | | |

The fitted linear results in Table 8, including humidity, temperature, and CN ratio of litter and organic matter have a considerable impact on total N, and the highest contribution rate. The impact of litter N content and pH value is slightly lower, but the *p*-values for all other influencing factors, with the exception of litter N content, are less than 0.05, indicating a significant result.

**Table 8.** Multivariate analysis of variance of soil ammonium nitrogen in *P. mongolica* forest (*t*-test).

| Model | Nonstandard Coefficient Beta | | Standardized Number Beta | *t* | Significance |
|---|---|---|---|---|---|
| Particular year | 0.04 | 0.02 | 0.59 | 2.75 | 0.05 |
| Forest type | 0.02 | 0.01 | 0.12 | 2.54 | 0.02 |
| Soil depth | 0.01 | 0.01 | −0.11 | −2.82 | 0.07 |
| pH value | 0.02 | 0.02 | 0.09 | 1.11 | 0.27 |
| Humidity | 0.03 | 0.01 | 0.53 | 4.81 | 0.01 |
| Temperature | 0.00 | 0.00 | −0.03 | −1.23 | 0.02 |
| Litter N content | 0.01 | 0.05 | 0.05 | 0.25 | 0.80 |
| Litter C content | −0.01 | 0.00 | −1.65 | −2.32 | 0.02 |
| Litter CN ratio | 0.03 | 0.01 | 2.39 | 2.66 | 0.01 |
| Soil organic matter content | 0.00 | 0.00 | 0.16 | 2.33 | 0.02 |

*3.4. Characteristics in Seasonal Dynamic Variation of Soil $NO_3^-$—N Content with Soil Depth and Stand Age*

During March and November, the soil $NO_3^-$—N content across the three forest stand ages first decreased and then increased with increasing soil depth (Table 4 and Figure 3). There were significant differences in $NO_3^-$—N content between the 0–10 cm, 10–20 cm, and 20–30 cm, 30–40 cm, and 40–50 cm soil layers (*p* < 0.05).In March, the maximum values for $NO_3^-$—N across the three forest ages were recorded from the 0–10 cm soil layer. The minimum values were recorded from the 30–40 cm soil layer. The average soil $NO_3^-$—N contents recorded in the near-mature, middle-aged, and young forests were 14.23 mg/kg, 13.58 mg/kg, and 12.64 mg/kg, respectively. In November, the maximum values of $NO_3^-$—N across all three forest age groups were recorded from the 0–20 cm soil layer, and the minimum values were recorded from the 30–50 cm soil layer. The average soil $NO_3^-$—N contents recorded from the near-mature, middle-aged, and young forests were 22.86 mg/kg, 14.83 mg/kg, and 14.23 mg/kg, respectively.

The soil $NO_3^-$—N content across the three forest ages during May, July, and September first increased and then decreased with increasing soil depth. There was a significant difference in the total N content between the 0–10 cm, 10–20 cm, 20–30 cm, 30–40 cm, and 40–50 cm soil layers (*p* < 0.05). During May, the maximum values of $NO_3^-$—N from young and near-mature forests were recorded from the 10–20 cm soil layer. The maximum value of $NO_3^-$—N from the middle-aged forest was recorded from the 20–30 cm soil layer. The minimum value of $NO_3^-$—N from the three forest age groups was recorded from the 30–50 cm soil layer.The average soil $NO_3^-$N contents recorded from near-mature, middle-aged, and young forests were 34.99 mg/kg, 29.14 mg/kg, and 27.04 mg/kg, respectively. In July, the maximum values of $NO_3^-$—N from young and near-mature forests were recorded from the 10–20 cm soil layer. The maximum value of $NO_3^-$—N from the middle-aged forest was recorded from the 20–30 cm soil layer. The minimum value of $NO_3^-$—N fromthe three forest age groups was recorded from the 30–50 cm soil layer. The average soil $NO_3^-$—N content recorded for young, middle-aged, and near-mature forests was 35.20 mg/kg, 29.83 mg/kg, and 25.18 mg/kg, respectively. In September, the maximum values for $NO_3^-$—N in young and middle-aged forests were recorded from the 0–20 cm soil layer. The maximum value for $NO_3^-$—N in near-mature forests was recorded from the 20–30 cm soil layer. The minimum value of $NO_3^-$—N from the three forest age groups was recorded from the 30–50 cm soil layer. The average soil $NO_3^-$—N content recorded for middle-aged, near-mature, and young forests was 36.85 mg/kg, 29.37 mg/kg, and 24.83 mg/kg, respectively.

In 2017, the soil $NO_3^-$—N increased first and then decreased over time. In May and July, the total N content for the three forest age groups was the highest recorded during the study.

By recording the sampling time, soil depth, forest type, pH value, soil temperature, soil humidity, soil organic matter, litter N content, litter carbon content, and litter C/N content as independent variables, the fitted linear regression model for all N was constructed. From the fitted linear results in Table 9, the *p*-value was significant at 0.001. The results are shown in Table 10.

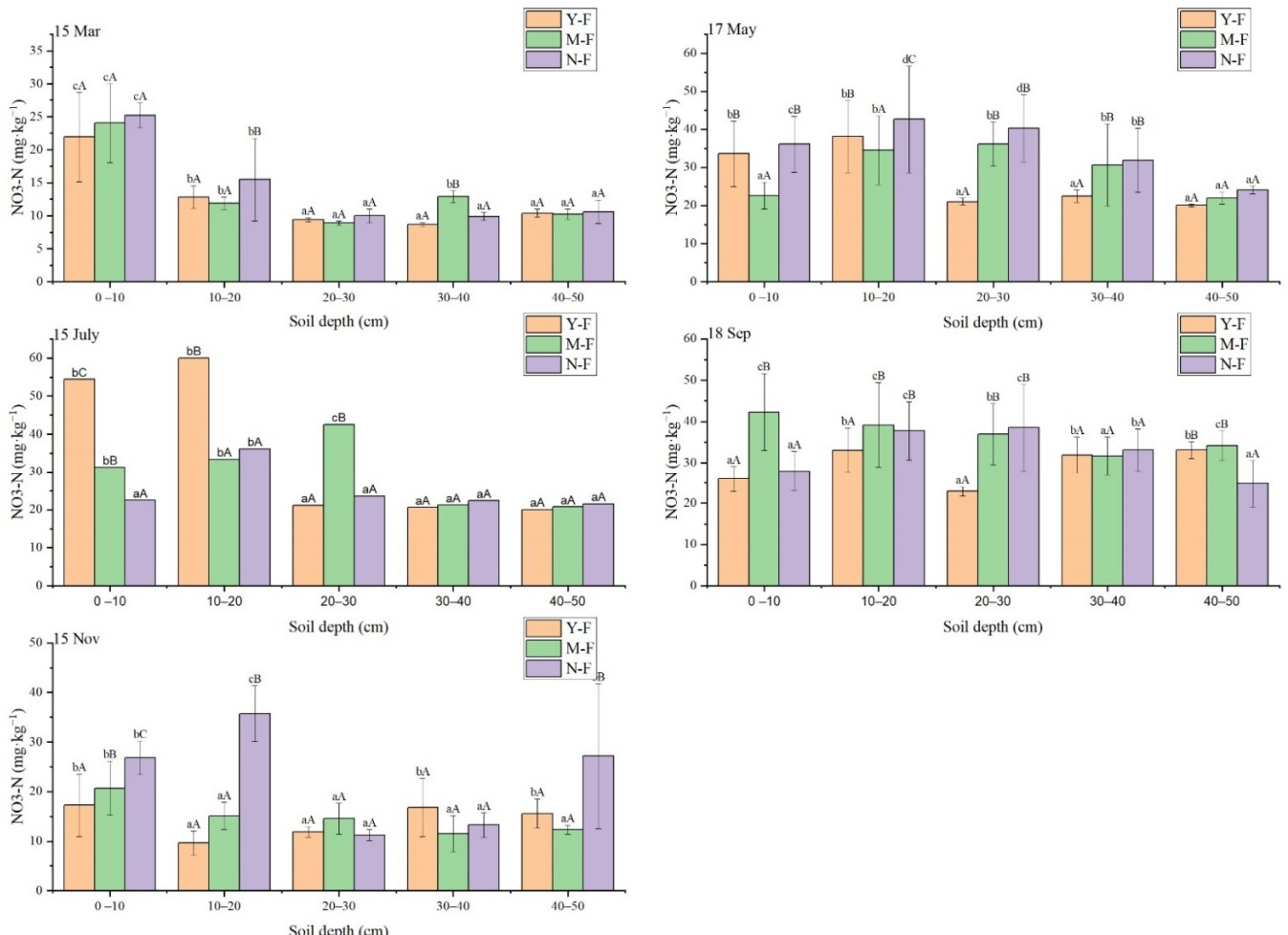

**Figure 3.** Plot of soil $NO_3^-$—N variation with soil depth. Lowercase letters represent the significant difference of different soil layers in the same forest age, and uppercase letters represent the significant difference of different forest ages in the same soil layer.

**Table 9.** Fittedlinear table for soil nitrate nitrogen in *P. mongolica* forest (*F*-test). The dependent variable was nitrate (N).

| Model | | Sum of Squares | Freedom | Mean Square | F | Significance |
|---|---|---|---|---|---|---|
| 1 | Regression | 1,348,848.91 | 10 | 134,884.89 | 2.98 | 0.001 |
| | Residual | 11,727,318.02 | 259 | 45,279.22 | | |
| | Total | 13,076,166.93 | 269 | | | |

From the fitted linear results in Table 10, humidity, temperature, and CN ratio of litter and organic matter have a considerable impact on the total N, and the highest contribution rate. The impact of litter N content and pH value is slightly lower, but the *p*-values for all other influencing factors, with the exception of litter N content, are less than 0.05, indicating a significant result.

**Table 10.** Multivariate analysis of variance of soil nitrate nitrogen in the *P. mongolica* forest (*t*-Test).

| Model | Nonstandard Coefficient Beta | | Standardized Number Beta | *t* | Significance |
|---|---|---|---|---|---|
| Sampling time | 13.24 | 47.05 | 0.10 | 0.28 | 0.08 |
| Forest type | 4.79 | 22.44 | 0.02 | 0.21 | 0.08 |
| Soil depth | 1.74 | 1.01 | 0.11 | 1.72 | 0.08 |
| pH value | −39.97 | 41.03 | −0.08 | −0.97 | 0.33 |
| Humidity | 54.79 | 15.91 | 0.41 | 3.44 | 0.01 |
| Temperature | −2.95 | 3.22 | −0.14 | −0.92 | 0.03 |
| Litter N content | −30.05 | 122.53 | −0.05 | −0.25 | 0.08 |
| Litter C content | −3.92 | 6.76 | −0.45 | −0.58 | 0.06 |
| Litter CN ratio | 12.71 | 23.19 | 0.54 | 0.55 | 0.05 |
| Soil organic matter content | −0.29 | 0.27 | −0.08 | −1.06 | 0.02 |

### 3.5. Correlation between Soil Total N, $NH_4^+$—N, $NO_3^-$—N Content and Month, Soil Depth, and Forest Age

The total soil N in the *P. mongolica* forest has a positive correlation with the sampling time, with a correlation coefficient of 0.394, a significant negative correlation with litter C content and litter CN ratio, with correlation coefficients of −0.120 and −0.129, respectively, and a positive correlation with ammonium N, temperature, humidity, litter N content, and soil organic matter content (Table 11). However, there is no significant correlation with soil depth, nitrate N and pH value also show a negative correlation. $NH_4^+$—N had a significant negative correlation with soil depth and pH value, and the correlation coefficients were 0.228 and −0.222, respectively. There was a significant positive correlation with forest type, temperature, humidity, and soil organic matter content, and the correlation coefficients were 0.47, 0.236, 0.388, and 0.264, respectively. There was little positive correlation with years, but with nitrate N, litter C content, and litter N content, the litter CN rate showed a negative correlation. $NO_3^-$—N had a positive correlation with year, temperature, and humidity, with clear significance. The correlation coefficients were 0.44, 0.157, and 0.265, respectively. There was a significant negative correlation with pH, with a correlation coefficient of −0.140. There was a weak positive correlation with soil depth, forest stand, and litter N content, and a weak negative correlation with total N, ammonium N, litter C content and litter CN rate.

**Table 11.** Correlation coefficient between nitrogen stoichiometry in the soil of *P. mongolica* forest and influencing factors * at level 0.05 (two-tailed) with a significant correlation. ** At the 0.01 level (two-tailed), the correlation was significant.

| | Sampling Time | Soil Depth | Forest Type | Total N | NNH₄N | NNO₃N | pH | Temperature | Humidity | Litter C Content | Litter N Content | Litter C/N | Soil Organic Matter Content |
|---|---|---|---|---|---|---|---|---|---|---|---|---|---|
| Sampling time | 1 | 0 | 0 | 0.394 ** | 0.11 | 0.44 ** | −0.164 ** | 0.251 ** | 0.258 ** | −0.957 ** | 0.919 ** | −0.973 ** | 0.052 |
| Soil depth | | 1 | 0 | −0.001 | −0.228 ** | 0.06 | 0.248 ** | −0.057 | −0.165 ** | −0.76 | −0.62 | −0.71 | −0.359 ** |
| Forest type | | | 1 | 0.21 | 0.47 ** | 0.006 | −0.285 ** | 0 | −0.047 | −0.196 ** | 0.124 * | −0.158 ** | 0.166 ** |
| Total N | | | | 1 | 0.037 | −0.005 | −0.029 | 0.078 | 0.008 | −0.120 * | 0.093 | −0.129 * | 0.066 |
| NNH₄N | | | | | 1 | −0.037 | −0.222 ** | 0.236 ** | 0.388 ** | −0.002 | −0.013 | −0.005 | 0.264 ** |
| NNO₃N | | | | | | 1 | −0.140 * | 0.157 ** | 0.265 ** | −0.038 | 0.024 | −0.044 | −0.012 |
| pH value | | | | | | | 1 | −0.246 ** | −0.413 ** | 0.237 ** | −0.280 ** | 0.229 ** | −0.531 ** |
| Temperature | | | | | | | | 1 | 0.819 ** | −0.185 ** | 0.072 | −0.266 ** | 0.205 ** |
| Humidity | | | | | | | | | 1 | −0.197 ** | 0.130 * | −0.255 ** | 0.233 ** |
| Litter C content | | | | | | | | | | 1 | −0.941 ** | 0.993 ** | −0.086 |
| Litter N content | | | | | | | | | | | 1 | −0.934 ** | 0.119 |
| Litter C/N | | | | | | | | | | | | 1 | −0.085 |
| Soil organic matter content | | | | | | | | | | | | | 1 |

## 4. Discussion

### 4.1. Effects of Litter Addition on the Contents of Total N, $NH_4^+$—N, and $NO_3^-$—N in the Soil of *P. mongolica* Forest

Forest litter is the main source of soil nutrients, and nutrients are returned to the soil following decomposition [3]. This study found that there was a positive correlation between litter N content and soil organic matter, total N content, and $NO_3^-$—N content across different forest ages. High N content in the litter and high soil organic matter, total N, and $NO_3^-$—N contents were observed. There was a negative correlation between litter N and $NH_4^+$—N contents. A negative correlation between litter C content and soil organic matter, total N, and $NO_3^-$—N contents was also observed. In this study, the total N and $NO_3^-$—N increased with the increase in N content during litter decomposition. The results are similar

to those of the Harvard Forest pine plantation in Petersham, Massachusetts, USA [15] and the Masson pine plantation in Hengxian Town, Nanning, Guangxi, China [4]. It is possible that N input increases the content of mineral N in the soil and litter layers, buffering the competition between plant absorption and nitrobacteria, as well as denitrifying bacteria for N, increasing nitrification and denitrification, and then increasing soil-available N [17]. The results showed that *P. mongolica* forest soil nitrification originated from ammoniated $NH_4^+$-N, and the change of nitrification rate was directly affected by ammonion. The nitrification rate was usually lower than the ammonion rate, and the promotion effect of nitrogen addition excitation effect on $NH_4^+$-N was higher than that of $NO_3^-$-N. The effect of nitrogen addition on net nitrification rate was not obvious [18].This may also be related to the mineralization and fixation of N. The addition of N improves the soil nitrification process, resulting in more N in the form of $NO_3^-$—N [19]. As the increased N is absorbed by soil organic matter, C/N decreases, which improves the release rate of N during decomposition [20]. It is also possible that the added inorganic N is fixed by microorganisms, which promotes the mineralization and release of the original organic N [21].

*4.2. Effects of Environmental Factors on the Contents of Total N, $NH_4^+$—N, and $NO_3^-$—N the Soil of P. mongolica Forest*

The change in soil N content is affected by various environmental factors such as soil temperature, moisture, and pH. Differences in temperature, humidity, and litter supply in different niches affect N mineralization by influencing the number, species, and vitality of different microbial groups in the forest [20].

In this study, a positive correlation was observed between temperature and soil total N, $NH_4^+$—N, and $NO_3^-$—N. The contents of total N, $NH_4^+$—N, and $NO_3^-$—N increased with increasing temperature. This was similar to the research results of soil nitrogen mineralization in karst native tree forest studied by Zhao et al. [18]. The monthly dynamic change of soil available nitrogen is the result of *P. mongolica* growth and soil microbial activity, because both are controlled by temperature and moisture. With the increase in temperature and humidity within a certain threshold, the microbial and enzyme activities were higher, and the decomposition of litters was faster, which accelerated the nitrogen mineralization process and increased the soil available nitrogen contents [18]. The contents of total N, $NH_4^+$—N, and $NO_3^-$—N on the soil surface of young, middle-aged, and near-mature forests were all the highest in July, and the lowest in November. This may be because the increase in temperature increases the availability of soil ammonia N and nitrate N, as the $NH_4^+$—N of forest soil ammonification is the source of nitrification, with ammonification directly affecting the change in nitrification rate. The nitrification rate is often lower than the ammonion rate [22]. The increase in temperature promotes the denitrification of the surface soil. In addition, when the temperature increases, the microbial growth and metabolism activity is enhanced and a large amount of organic matter is decomposed. This improves the mineralization rate of soil N and significantly increases the content of N in the soil [23]. Temperature change can also change the mineralization rate of N in the soil by affecting soil water content [24].

Soil moisture content is an important factor in the process of soil N transformation. In this study, the soil total N, $NH_4^+$—N, and $NO_3^-$—N were positively correlated with soil moisture, and the contents of $NH_4^+$—N and $NO_3^-$—N were significantly correlated with soil moisture. This may be because the joint action of soil water content and other soil physical and chemical properties can significantly alter the porosity and pore distribution of soil. This affects the circulation of oxygen in soil, which in turn affects the activity of microorganisms [24]. The region has a short summer (July–August) with high temperature and high humidity. The short-term increase in temperature and water can significantly improve the activity of soil microorganisms, which is conducive to their growth and reproduction. This can change the contents of soil total N, $NH_4^+$—N, and $NO_3^-$—N. As drought and low temperatures weaken biological activities, the litter decomposition rate

decreases to a certain extent with low temperatures during winter (November) and during low precipitation (May).

Soil pH and other pH can directly or indirectly affect other properties and are the main variables affecting soils [25,26]. In this study, soil pH was negatively correlated with soil total N, $NH_4^+$—N, and $NO_3^-$—N, as well as with $NH_4^+$—N and $NO_3^-$—N. This is similar to the research results of a moist evergreen broad-leaved forest of WawuMountain by Chen et al. [27]. Lower pH will limit the growth of soil denitrifying microorganisms. However, lower pH may reduce the availability of organic carbon and mineral N available to denitrifying microorganisms [28].

*4.3. Effects of Seasonal Variation on the Contents of total N, $NH_4^+$—N, and $NO_3^-$—N in the Soil of P. mongolica Forest*

In the current study, the total N content of *P. mongolica* forest across three different stand ages increased first and then decreased over time during 2017. The total N content for the three forest ages was the highest during May and July. The content of $NH_4^+$—N across the three forest ages and in each soil layer is greater than that of $NO_3^-$—N, which is consistent with the research results of decomposition of leaf litter of *Picea crassifolia* Forest in the Qilian Mountains [11]. This is because the N element mainly exists in the form of organic matter, and its release needs to be decomposed by microorganisms. In summer, microbial activity begins to increase, which promotes the decomposition of the N element [29]. This indicates that $NH_4^+$—N is the main form of soil available N. Correlation analysis showed that seasonal dynamic changes had an effect on the $NH_4^+$—N and $NO_3^-$—N in the *P. mongolica* soil layer. The content was lower than in March and November, and higher during May and July [3]. This may be because the weather warmed up during May, the snow melted, the soil temperature and humidity increased at the same time, the soil microbial activity began to increase, and the N mineralization, especially the ammoniation, increased [17]. This resulted in a large amount of decomposition of the N in the litter, and the contents of $NH_4^+$—N and $NO_3^-$—N in *P. mongolica* forest soil for each forest age would have increased. The *P. mongolica* then entered the growing season, and with the continuous increase in temperature, it needs to absorb a large amount of $NH_4^+$—N [30,31]. Part of the $NH_4^+$—N is transformed into $NO_3^-$—N through the action of nitrifying microorganisms. With the increase in rainfall, $NH_4^+$—N and $NO_3^-$—N entered the deep soil layer through the leaching of rainwater. The $NH_4^+$—N content then decreases to a certain extent. The high content during July was due to the slow growth of *P. mongolica* during autumn. The reduction in $NH_4^+$—N absorption and its relative accumulation, is consistent with the results of $NH_4^+$—N and $NO_3^-$—N in temperate forest soils studied by Xu et al. [3]. Zhao et al. [18] showed that soil with a low temperature in winter still had an obvious nitrogen mineralization process. In November, the contents of $NH_4^+$—N and $NO_3^-$—N in *P. mongolica* soils were 0–20 cm. The contents of the soil surface layer were greater than that at depths of 30–50 cm. The main reason for this is that the northern temperate zone enters winter in October, the weather is cold, and the soil is covered with ice and snow. Low temperatures will inhibit the activities of soil microorganisms, weaken the humic effect and slow decomposition rates, which hinders the mineralization of soil [30]. In addition, winter is the non-growing season, and the *P. mongolica* needs less nitrogen, and the available nitrogen is abundant in the soil.

*4.4. Effect of Soil Depth on the Contents of Total N, $NH_4^+$—N, and $NO_3^-$—N in the Soil of P. mongolica Forest*

Results showed that the total N, $NH_4^+$—N, and $NO_3^-$—N in *P. mongolica* soils decreased with increasing soil depth (0–50 cm), There were significant differences between 0–10 cm, 10–20 cm, 20–30 cm, 30–40 cm, and 40–50 cm soil layers ($p < 0.05$). This is consistent with trends in soil N content with increasing soil depth in forests in the Qinghai Province as well as the alpine forests in western Sichuan [31,32]. This is because the process of decomposition and synthesis of litter returns N to the soil. On the soil surface, the N

content mainly comes from litter decomposition. While litter is mainly concentrated on the soil surface, nutrients also accumulate there. Good ventilation and hydrothermal conditions on the soil surface provide a better environment for microbial activities, therefore promoting the accumulation of N content at the soil surface. Then the N migrates and diffuses down to the mineral soil layer with water or other media. In the deeper soil layers, the N content is mainly derived from roots, root exudates, soil microorganisms, and N leaching from some of the upper layers. Compared with the soil surface layer, the exchange with the outside world is weak [33]. In addition, with the deepening of the soil layer, plant roots, soil animals, and microorganisms absorb and utilize nutrients. As a result, the total N, $NH_4^+$—N, and $NO_3^-$—N in *P. mongolica* soils decreased with the increase in soil depth (0–50 cm). This is consistent with the results of Qin et al. [34] on soil nutrients in different forest types with Masson pine (*Pinus massoniana* Siebold and Zucc.).

*4.5. Effects of Forest Age on the Contents of Total N, $NH_4^+$—N, and $NO_3^-$—N in the Soil of P. mongolica Forest*

This study has shown that the content of soil $NH_4^+$—N during March and November was significantly different from that of May and August ($p < 0.05$).The $NH_4^+$—N content in the soil of the *P. mongolica* forest generally showed a trend of near-mature forest > middle-aged forest > young forest. The trend in soil $NO_3^-$—N content is consistent with that of *P. mongolica* soil $NH_4^+$—N content. This is similar to the resultson *Larch plantation* [35] and *P. massoniana* plantation [36] in different years. Itmay be that during the growth of the *P. mongolica* forest, the *P. massoniana* at different growth stages have different nutrient needs, which can lead to differences in nutrient content in their own organs and litter. At the stage of *P. mongolica* young forest to middle-aged forest, there was agradual increase in biomass and canopy density of *P. mongolica* forest, decrease in understory vegetation and water and deficiency in understory light. As the *P. mongolica* entered into the mature stage, all the conditions became better by self-thinning, natural pruning, etc. The light conditions under the forest were improved. The vegetation under the forest developed rapidly. The soil surface litter and animal and plant residues gradually increased, the soil texture has been greatly improved, and the number and variety of microorganisms are various and active. They decompose the soil surface litter and animal and plant residues, resulting in an increase in soil nutrients $NH_4^+$—N and $NO_3^-$—N [4,37,38]. Due to the demand for N in different growth stages of *P. mongolica*, the demand for soil N in young and middle-aged forest stages is high. After the forest has reached maturity, due to the slow and stable growth of *P. mongolica*, the utilization rate of soil N may be lower. In addition, the deeper the soil has roots growing in the near-mature forest of *P. mongolica*, the more conducive it is to the absorption of deep-seated soil nutrients. This also leads to the highest contents of $NH_4^+$—N and $NO_3^-$—N at the soil surface when the forest reaches maturity [35].

**5. Conclusions**

The nutrient release of *P. mongolica* litter was affected by decomposition time in three forest ages (young, middle-aged, and near-mature forest). The content of C and C/N in litters of young, middle-aged, and near-mature forests decreased with the increase in decomposition time. There was significantly different C content of litters at different ages ($p < 0.05$). N content in litters increased with time. There was significantly different N content of litter at different ages (except March) ($p < 0.05$). The effects of litter addition on total N, $NH_4^+$—N, and $NO_3^-$—N in *P. mongolica* soil were positively correlated. The changes of total N and $NH_4^+$—N contents in *P. mongolica* forest were as follows: near-mature forest > middle-age forest > young forest. The contents of $NH_4^+$—N and $NO_3^-$—N in *P. mongolica* forest were affected by seasonal dynamics and soil temperature and humidity. The content was lower in March and November, and higher in May and September. $NH_4^+$—N in all soil layers of *P. mongolica* was greater thanNO$_3^-$—N in the same month. Total N, $NH_4^+$—N, and $NO_3^-$—N decreased with the increase in soil depth (0–50 cm). There was significantly different $NO_3^-$—N content ($p < 0.05$), while there was no signif-

icantly different in $NH_4^+$—N content between middle-age forest and near mature forest with the increase in soil depth.

In this study, we focus on the decomposition of fresh leaf litter and the release of some nutrient elements in *P. mongolica* soil within one year, and the effects of fresh leaf litter decomposition on N elements in *P. mongolica* soil of different forest ages has been analyzed. Due to the environment in the agro-pastoral ecotone in northern China and the interaction between the environment and litter-soil of different ages of *P. mongolica* forest, soil nutrients are the key factors of the decomposition environment of *P. mongolica* litter-soil. The interaction between soil nutrients and the decomposition characteristics of *P. mongolica* litter-soil is diverse and complex. The changes of leaf structure and the relationship between quantification and habitat factors of *P. mongolica* leaf litter need further study in the later stage.

**Author Contributions:** Conceptualization, L.C. and Y.L.; methodology, Y.L.; validation, Y.L., L.C. and W.D.; formal analysis, X.L.; investigation, Y.B. and Y.L.; resources, Y.L., L.C., W.D. and Y.B.; data curation, Y.L. and X.L.; writing—original draft preparation, Y.L.; writing—review and editing, Y.L., L.C. and W.D.; visualization, Y.L., L.C. and W.D.; supervision, W.D.; project administration, Y.L. and Y.B. All authors have read and agreed to the published version of the manuscript.

**Funding:** This research was funded by The Natural Science Foundation of Inner Mongolia (Project 2017MS0327),and The Inner Mongolia autonomous region of institutions of higher learning scientific research projects (Project NJZZ22176).

**Data Availability Statement:** Not applicable.

**Acknowledgments:** We are grateful to friends who assisted with fieldwork, especially Yichao Wang. We thank the Baiyinaobao National Nature Reserve Management Administration staff for enabling this research.

**Conflicts of Interest:** The authors declare no conflict of interest.

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
