# Peer review of "Effects of Litter Decomposition on Soil N in Picea mongolica Forest at Different Forest Ages"

_forests, doi:10.3390/f13040520_

Round 1

Reviewer 1 Report

Title

You can take under consideration deleting the double use of the word "forest"

Abstract

I think that a specific research aim should be introduced in the abstract (at least as in the Introduction) and not only the research results and what was the subject of the research.

Introduction

This chapter is the weakness of this work. The chapter is very shortened. There are few specifics, many general terms. It should be more specific. I do not consider it necessary to introduce this statement "The results from this study can provide a reference for use in the management and cultivation of artificial forests."

Materials and Methods

This chapter is described reasonably well. However, certain rules should be maintained (this also applies to other chapters). A detailed comment was also indicated in the manuscript. a) The text should refer to the inserted tables b) We have to use only three real numbers in the tables, text. Tables are readable there.

Results

This chapter is relatively well written and is the strongest point of the manuscript. However, it should be based on substantively proven data (see manuscript and table 3) Discussion This chapter does not need to be corrected Conclusion

The conclusions are specific and relevant

References

Well chosen

Round 2

Reviewer 2 Report

I am glad that the authors have addressed my comments and concerns during the first review. However, I am still not convinced with what the authors changed in the results section. It needs more trimming and focus. Please do not put everything in the results paragraph that has been presented already in the tables or figures. I also suggested some figures that do not necessarily help explain the results. Please see my comments below:

Line 201 : ‘mouth’ into ‘month’?

Lines 224-225: ‘The minimum age across the three forest age groups was recorded in the 30–50 cm soil layer’. The authors talked about ‘age’, which age are they referring to?

Figures 2 and 3, 5, and 6, 8, 9, and 10  only increased the confusion and were not even discussed what these figures represent? I suggest removing them from the manuscript if they do not necessarily explain much about the results. They do not provide substantial value to discuss the results. Focus more on important highlights of your result and go deep into it, which you did well in the discussion section.   

Table 5: I suggest you remove the non-standard coefficient values. It only adds up the width of the table. Else, make a similar table like in Table 6. Put asterisks according to the level of significance in the ‘Significance’ column.

Figures 4 and 7, put asterisks for significant differences and put error bars. Why the x-axis title is at the edge? Please put it in the middle below the x-axis labels.   

Line 315 : is this ‘greater’ or ‘lesser’?

Discussion: The Discussion section has substantially improved. Hopefully, the results section will be as impressive as the Discussion section.

General: Have an English editor proofread the manuscript to make it nicely written. The authors do not have to describe every single detail of what was written in the tables. These only lengthen the manuscript with not much substance. Only highlight the most important ones.

Author Response

Re: forests-1553653

Dear Ms. Denise Li

Thank you very much for returning the reviewers’ comments on our manuscript (forests-1553653) entitled “Effects of litter decomposition on soil N in Picea mongolica forest at different forest ages”. All comments are valuable and helpful for improving the quality of the paper. We have fully considered the reviewers’ comments and thoroughly revised the paper. We hope you will be satisfied with our revision. We also appreciate your great efforts in improving our paper. Based on the reviewers’ suggestions, the manuscript has been revised. In the following, we outline the associated changes and/or responses in [See Line No.], and we explain why minor comments cannot be accommodated.

Yours sincerely,

Lixin Chen

Responses to the comments by Editor:

  1. Figures 2 and 3, 5, and 6, 8, 9, and 10 only increased the confusion and were not even discussed what these figures represent? I suggest removing them from the manuscript if they do not necessarily explain much about the results.

Response: Thank you for reminder. We have removed them from the manuscript.

  1. The Discussion section has substantially improved. Hopefully, the results section will be as impressive as the Discussion section.

Response: Thank you for reminder. We have improved discussion section。please refer to the Line373-Line378,Line394-Line401,Line465-Line466,Line497-Line503,

  1. Minor comments

Response: Thank you for reminder. All the questions have been modified in the article. Thank you very much for your valuable comments.

We deeply appreciate your valuable suggestions, and we look forward to receiving comments from reviewer. Again, we really appreciate your comments that are very helpful in improving the quality of our paper.

Sincerely yours,

Lixin Chen     

College of Forestry      

Northeast Forestry University
